# Threshold Voltage Degradation for n-Channel 4H-SiC Power MOSFETs

**Esteban Guevara [1], Victor Herrera-Pérez [1],***, **Cristian Rocha [2] and Katherine Guerrero [2]**

[1]  Facultad de Informática y Electrónica, Escuela Superior Politécnica del Chimborazo (ESPOCH), Riobamba 060150, Ecuador; esteban.guevara@espoch.edu.ec

[2]  Sistemas de Comunicaciones y Electrónica Aplicada (SICOMEL), Universidad Nacional de Chimborazo (UNACH), Riobamba 060151, Ecuador; crocha@unach.edu.ec (C.R.); kaguerrero@unach.edu.ec (K.G.)

*  Correspondence: isaac.herrera@espoch.edu.ec; Tel.: +593-987334746

**Abstract:** In this study, threshold voltage instability on commercial silicon carbide (SiC) power metal oxide semiconductor field electric transistor MOSFETs was evaluated using devices manufactured from two different manufacturers. The characterization process included PBTI (positive bias temperature instability) and pulsed IV measurements of devices to determine electrical parameters' degradations. This work proposes an experimental procedure to characterize silicon carbide (SiC) power MOSFETs following two characterization methods: (1) Using the one spot drop down (OSDD) measurement technique to assess the threshold voltage explains temperature dependence when used on devices while they are subjected to high temperatures and different gate voltage stresses. (2) Measurement data processing to obtain hysteresis characteristics variation and the damage effect over threshold voltage. Finally, based on the results, it was concluded that trapping charge does not cause damage on commercial devices due to reduced value of recovery voltage, when a negative small voltage is applied over a long stress time. The motivation of this research was to estimate the impact and importance of the bias temperature instability for the application fields of SiC power n-MOSFETs. The importance of this study lies in the identification of the aforementioned behavior where SiC power n-MOSFETs work together with complementary MOS (CMOS) circuits.

**Keywords:** silicon carbide MOSFET; pulsed IV measurements; stress modeling; hysteresis; threshold voltage; recovery voltage

---

## 1. Introduction

Silicon carbide (SiC) instead of silicon (Si) material is positioning itself as an alternative to manufactured MOSFETs, mainly by taking advantage of its high temperature operation stability, wide bandgap energy, high blocking voltage, ten times larger critical field, larger saturation velocity, and a greater thermal conductivity [1]. Besides, 4H-SiC is used to manufacture power MOSFETs and it is starting to become commercially available for power electronics applications [2]. Power MOSFETs manufactured by SiC will have smaller drift zones to those manufactured in silicon, with identical voltage and on-resistance $R_{ON}$. The used area can be reduced, allowing SiC MOSFETs to have one hundred times lower gate-source and gate drain capacitances [3,4]. SiC MOSFETs give significantly shortened dynamic and static losses, and they work at higher temperatures, higher power densities, and higher frequencies. These characteristics definitely incorporate system benefits.

For scaling down, the number of passive components of inverter integrated circuits' (ICs) additional heat sinks is reduced, accomplishing full silicon carbide-based system solutions which are much lightweight, more compressed, cheaper, and more efficient [5,6]. Our work studied the

threshold voltage instabilities of commercially available SiC MOSFETs by analyzing the positive bias temperature stress (PBTS) behavior, which is a reliability goal. PBTS causes trapping of carriers near the SiC/ $SiO_2$ interface, producing a variation on output characteristics from devices [2,7]. This work aimed to carry out a more comprehensive analysis of bias temperature instability (BTI), by evaluating and modeling voltage and temperature stress influence on the threshold voltage shift for devices under experimentation.

The threshold voltage variation $\Delta V_{th}$ occurs if a positive and negative gate polarization supply voltage is referred to its equivalent temperature instability due to the oxide trap. If the polarization voltage of the gate is removed, a recovery is generated that accelerates when the voltage goes in the opposite direction to the voltage [8].

The objective of this article is to highlight and help to better understand the variation of the threshold voltage of SiC power MOSFETs. Based on the idea that silicon carbide is a unique wide-bandgap (WBG) semiconductor and that it has a native oxide with high quality, the SiC/$SiO_2$ interface is distinct from the Si/$SiO_2$ interface due to the narrower band offsets to the dielectric, a wider band gap, vacancies inside the structure, and carbon associated point defects which only prevail in SiC [9–11]. The main contribution of this work was detecting the instability of the threshold voltage drifts that mainly affects to the reliability of the high powered devices and integrated systems. Due to the threshold voltage shift, the positive and negative bias temperature instability contributes to inhomogeneous current distributions in the system; for this reason the commutation degrades and the temperature in the module increases.

## 2. One Spot Drop Down (OSDD) Characterization Method

The one spot drop down (OSDD) characterization method is illustrated in Figure 1. BTI stress is fixed and $V_G$ (voltage gate) is reduced from $V_{G-STR}$ (voltage gate stress) to a convenient $V_{G-SNS}$ (voltage gate sense bias) to evaluate $I_{D-LIN}$ (linear drain current) in $t_M$ (measure time) spaced in logarithmic time intervals. The OSDD technique also suffers from recovery problems; nonetheless, it takes a much shorter time to measure a single spot drain current ($I_D$) than a full $I_D - V_{GS}$ sweep. Consequently, recovery can be reduced.

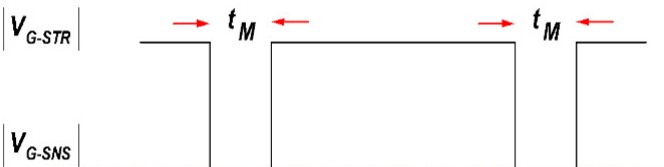

**Figure 1.** Schematic measured of OSDD technique employed.

As shown in Figure 2, once post-stress $\Delta ID_{LIN}$ is measured at $V_{G-SNS}$, it can be compared to pre-stress $I_D - V_{GS}$ sweep to determine BTI degradation. In the vertical shift method, Equation (1) can be estimated by noting the difference in $I_{D-LIN}$ between pre-stress and post-stress at $V_{G-SNS}$, and in the absence of mobility variation [12].

$$\Delta Vth = \frac{-I_{D-LIN}}{V_{G-SNS} - V_{T0}}. \tag{1}$$

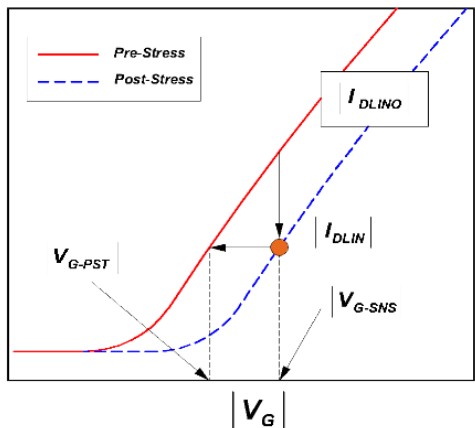

**Figure 2.** Measured $I_{DLIN}$ versus $V_G$ sweep before stress and one spot $I_{DLIN}$ measurement after stress.

### 3. Experimental Procedure and Measurements

A practical way to produce stress on MOSFETs suggests the application of high temperature gate bias (HTGB) where SiC power MOSFET degradation accelerates significantly. The objective of the study was to analyze the positive and negative bias temperature instability (PBTI and NPBTI) and transfer characteristics. The devices under scrutiny were two different n-channel SiC power MOSFET families provided by two different manufacturers. These devices were submitted to a thorough review that determined electrical parameters' degradations in the metal oxide semiconductor (MOS) structure; we used probes at a fixed temperature of 150 °C, mainly to measure experimental $I_D - V_{GS}$ characteristics in order to evaluate the trapping phenomena in the gate dielectric or in the interface between the gate dielectric and the SiC layer.

The characterization process was carried out by using the 4200-SCS Semiconductor Characterization System of Keithley Instruments, setting a suitable source measure unit (SMU), and setting a pulsed measure unit (PMU). The devices being tested were SiC power MOSFETs, characterized by a breakdown voltage of 1200 V and the drain-source on resistance $R_{DS_{ON}}$ at $V_{GS} = 20$ V equal to 52 m$\Omega$.

All measurements were carried out at high temperature, 150 °C, and with $V_{DS} = 50$ mV, to achieve low drain bias and to keep oxide field ($E_{OX}$) approximately constant across the channel during the application of stress [9]. The following measurement method consists of three main phases: the initial stabilization phase, the stress phase, and the recovery phase. The degradation in SiC power n-MOSFET drain current can be frequently checked in the gate stress. The initial stabilization phase is performed by applying a negative gate voltage: $-5$ V for 10 s. In this phase, the virgin device is stabilized by releasing the charges enclosed in trapping centers. The determination of the BTI-induced, parametric degradation of MOSFETs is frequently done by stressing the device at an accelerated aging condition, through a suitable gate bias ($V_G$) that is larger than ($V_G = V_{G-STR}$) than the one used during ordinary operation [13].

The experimental stress phase was performed at gate bias by varying it from 12 to 20 V at 150 °C. The accelerated stress test was performed at 1000 ms. After the stress phase, the recovery phase was performed by biasing the SiC power MOSFET with a voltage gate recovery ($V_{G-RECOVERY}$) ranging at intervals from 0 to $-2$ V for at least 1000 s.

Pulsed I–V measurements were carried out by means of a parameter analyzer: a Keithley 4200-SCS equipped with the 4225-PMU ultra-fast I–V module and two 4225-RPM remote amplifier/switch modules. I–V curves were achieved by the implementation of a train of pulses with a period of 300 μs and a width of 30 μs at the drain and gate of the device. The gate voltage pulse amplitude was varied from $-5$ to 20 V with a step of 100 mV.

Two different devices families provided by two different manufactures have been investigated. The fundamental electrical parameters of the devices are reported in Table 1.

**Table 1.** Main electrical parameters of the investigated devices.

| Device | Device 1 | Device 2 |
|---|---|---|
| BVdss [V] | 1200 | 1200 |
| Idsmax@25 °C [A] | 45 | 65 |
| Rdson typ@20A, $V_{GS}$ = 20 V, T = 25 °C [mΩ] | 80 | 52 |
| Vth@1mA [V] | 3.5 | 3 |
| Ciss@ (Vds = 50 mV ) [nC] | 2.5 | 3.5 |
| Temp max [°C] | 200 | 200 |
| Pack | HiP247$^{TM}$ | HiP247$^{TM}$ |

## 4. Results and Discussion

### 4.1. Bias Temperature Instability (BTI) in the SiC MOSFET Transitor

The BTI measurement procedure consists of three steps: initial stabilization, measurement of the ID–VGS reference curve, and multiple stress-sense measurements. Figure 3 depicts the progressions of ID–VGS curves with stress in the situation of each sample. Subsequently, at each stress period, the ID–VGS curves shifted toward the positive VGS direction, displaying clearly, a threshold voltage instability (similar behavior shown for the devices A and B).

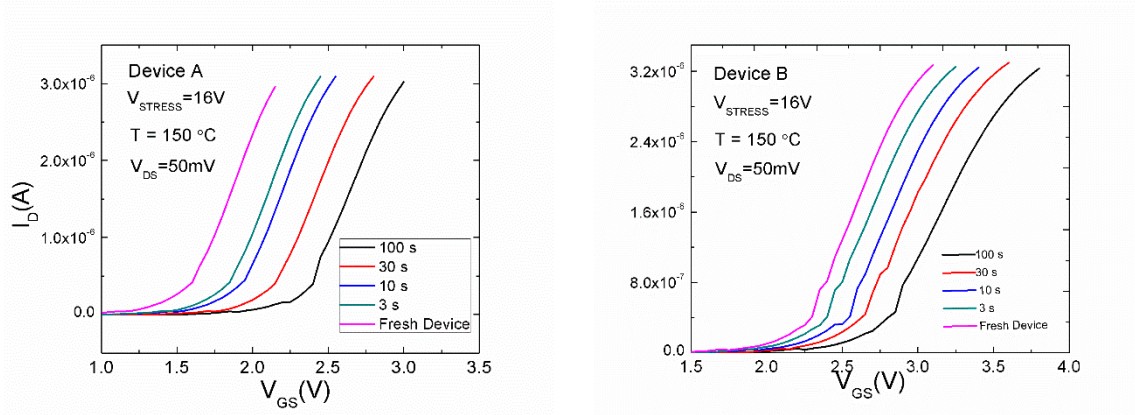

**Figure 3.** ID–VGS curves before (the initial stabilization phase was performed for each new sample) and after successive stress experiments for the devices A and B.

The initial stabilization phase is to achieve biasing a negative gate voltage of −5 V for 10 s. In this phase, the virgin sample is stabilized by releasing the charges originally trapped in the SiC/SiO$_2$ interface or in the bulk as shown in Figure 4. As result of the concomitant charge trapping and releasing during the stress, it was evaluated a $I_{DS} - V_{GS}$ curve sweeping $V_{GS}$ ranging from −5 to 3.5 V. It was necessary to measure the curve above 3 V, which is higher than $V_{th}$ [14]. Similar behavior was shown for Device A and Device B. In fact, after the recovery phase, the stress phase experiments for distinct stress voltages were performed by showing that the $I_{DS} - V_{GS}$ curve never comes back to this reference state phase.

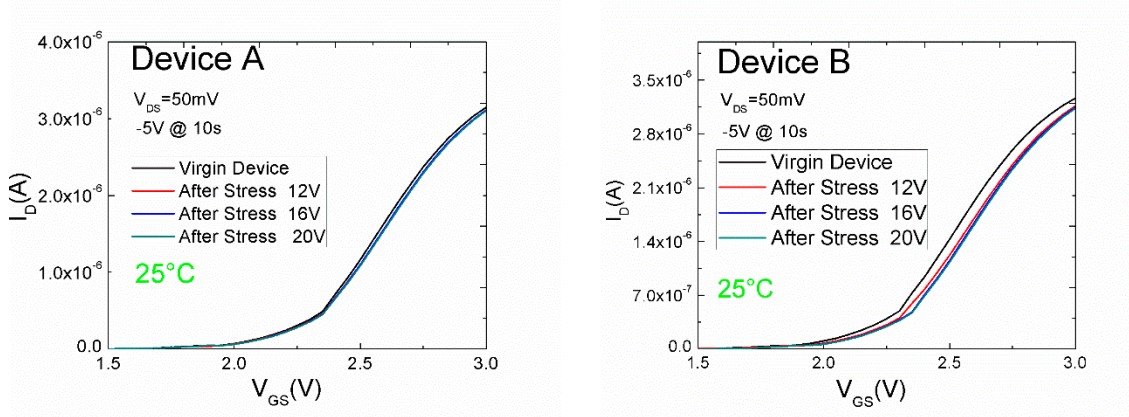

**Figure 4.** Transfer curves in the linear regime for a virgin device (device A and B), after the initial stabilization phase and after successive stress–recovery experiments.

The observed decrease and increase of the linear drain current at sense conditions is converted in the threshold voltage shift by using the $I_{DS} - V_{GS}$ reference curve, as depicted in Figure 5 (similar behavior is shown for both devices). The observed $\Delta V_{th}$ shift can be attributed to the electron trapping and de-trapping from the channel (under the gate oxide) into the traps placed in the $SiO_2$ energy gap (see Figure 6) [15,16].

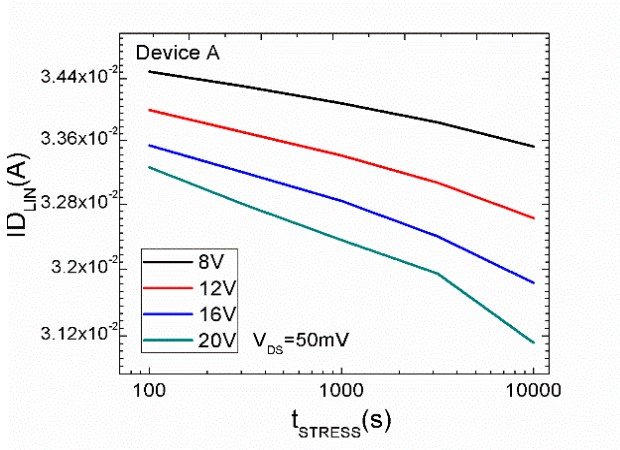

**Figure 5.** Evolution of the linear drain current versus stress time obtained from the test procedure during the stress phase.

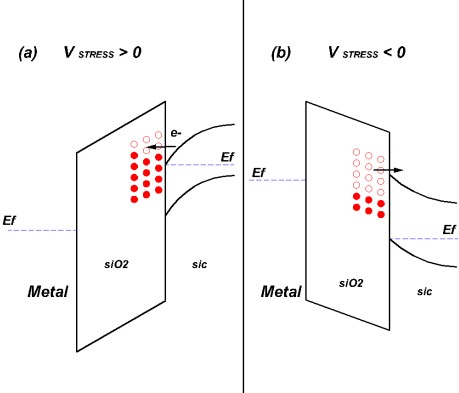

**Figure 6.** Representation of the sketch of the band diagram for the stress phase (**a**) and the recovery phase (**b**) obtained during the stress phase.

Figure 7 shows $\Delta V_{th}$ during the stress phase at diverse stress temperatures and voltages. In order to determine the $\Delta V_{th}$ evolution, the threshold degradation-time dependence was compared on both devices. Clearly, a good fit with the classic power law model is shown. Devices A and B showed the same $\Delta V_{th}$ in the gate bias regime when $V_{G-STR}$ adopted values of 12, 16, and 20 V at 150 °C within the measurement window from 0 to 1000 s. The stress conditions were interrupted at fixed time intervals and measure ID at $V_{G-SENSE} = 3$ V and compared with the initial reference curve (similar behavior for the devices A and B).

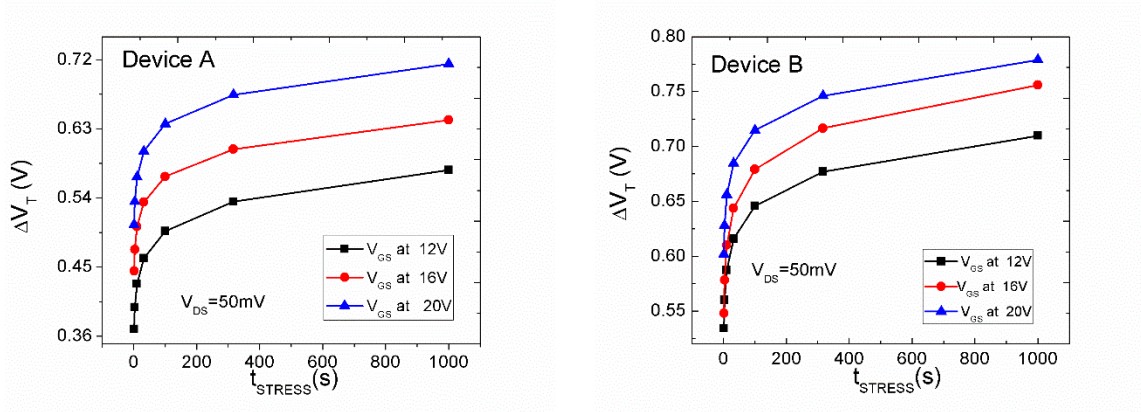

**Figure 7.** $\Delta V_{th}$ evolution during the stress phase at different stress voltages for the devices A and B.

As shown in Figure 8, the bias instability at different voltages does not show any temperature dependence (similar trends for devices A and B). Both examples exhibited similar time evolution of $\Delta V_{th}$ for both stress voltages. In all cases, a power law behavior with a slope of about 0.05 was observed.

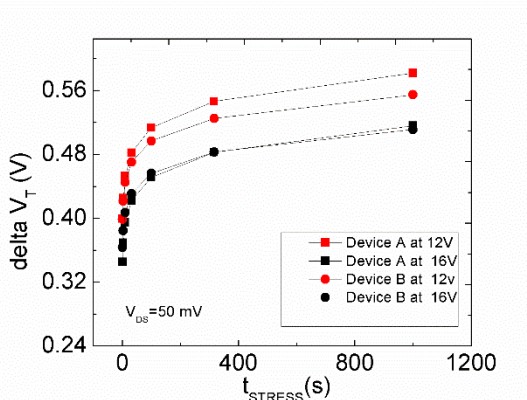

**Figure 8.** Time evolution of the threshold voltage shift $\Delta V_{th}$ induced by different gate voltages for Device A and Device B. A similar power law behavior was observed in the two types of SiC MOSFETs.

In the recovery phase, the suitable current level to return the drain current to the initial value is in the range of minutes, as shown in Figure 9. The observed $\Delta V_{th}$ evolution at the recovery phase is attributed to the release of electrons into the SiC layer from the energy trap states of the $SiO_2$ band gap. For the proposed experimental conditions, this behavior does not cause permanent damage in the test devices.

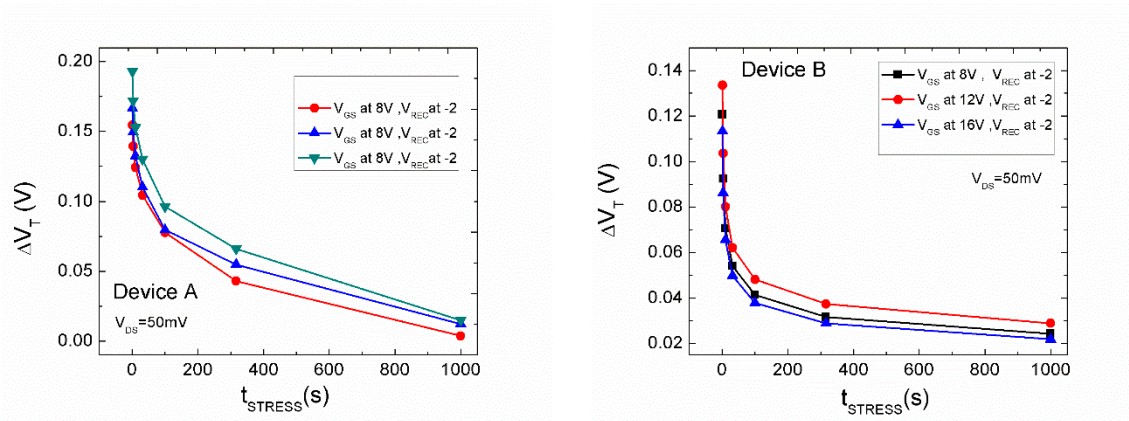

**Figure 9.** Time evolution of $\Delta V_T$ recovery measured at different $V_{G-REC}$ for the devices A and B.

Two different values of $V_{G-REC}$ have been used, as shown in Figure 10. Note that $\Delta V_{th}$ starts to recover immediately after stress when the devices are recovered at $V_{G-REC} = 0$ V. It demonstrates very a similar impact of recovery $V_{G-STRESS}$ for both devices, as shown in Figure 10.

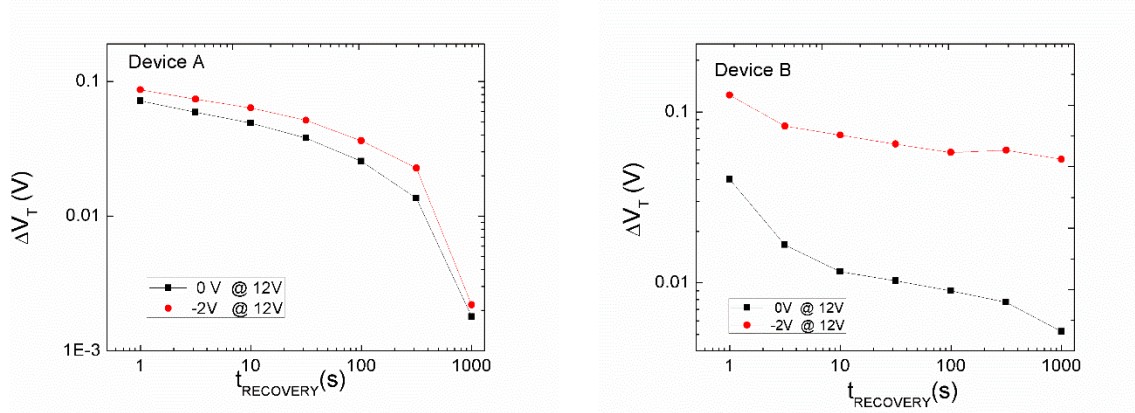

**Figure 10.** Recovery measurements at different voltages for the devices A and B.

### 4.2. Hysteresis between Two Adjacent Sweeps.

Considering the data depicted in Figure 11, the results show that both devices show some levels of defectiveness in their interface layers. The amplitude increases by raising the maximum gate voltage being applied, since it allows filling the traps at higher energy levels. Devices A and B show different drain current degradations at the same gate voltage. The degradation of current can be attributed to traps enclosed at the lateral regions or oxide/SiC interface, generating a change of the $\Delta V_{th}$. In order to quantify the hysteresis in the I–V curve, the maximum voltage shift $\Delta V_{MAX}$ was measured. The difference between $V_{TH-down}$ and $V_{TH-up}$ was named threshold voltage hysteresis ($\Delta V_{TH-HYST}$) [17,18]. The observed hysteresis between up-sweep and down-sweep can be expressed as a threshold voltage shift and may reach several millivolts. It was demonstrated that the threshold voltage shift is caused by electron capture produced by applying a gate bias in the $SiO_2$ and $SiC/SiO_2$ interface. Moreover, the inset shows that the ID–VGS overlaps in the initial ascending part, confirming that the starting bias at $-5$ V allows resetting the device characteristics by releasing the previously trapped charge. This observation also implies that the threshold voltage shift induced by the gate bias is recoverable, at least in the range of gate voltage investigated.

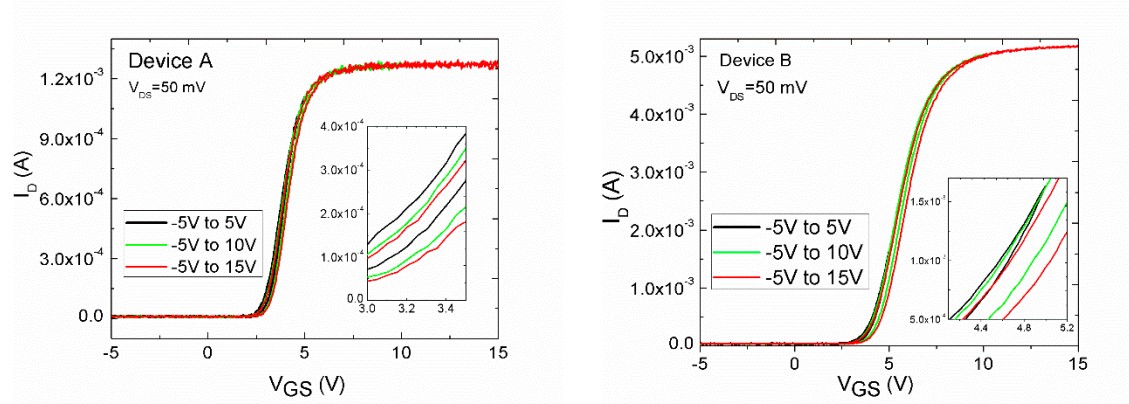

**Figure 11.** Hysteresis observed in the ID–VGS curve for the Device A and Device B. Inset highlights the increase of the hysteresis amplitude for raising maximum values of the gate voltage applied for the devices A and B.

The results shown above were obtained in the microelectronics laboratories of the University of Calabria. After the activity carried out by the two characterization methods mentioned before on a set of devices, a trend in the graphics could be found, without significant variations between devices.

Figure 12 shows the mean values of the hysteresis amplitude evaluated at a fixed current level (ID = 5 mA) against the mean values of $\Delta V_{th}$ (which were measured with the OSDD method) induced by a PBTI stress (temperature at 150 °C and stress time = 1 s). The two phenomena exhibit similar temperature dependence and are strongly correlated, suggesting that the same physical mechanism is the main factor responsible for hysteresis and PBTI. A clear correlation between hysteresis and PBTI was observed.

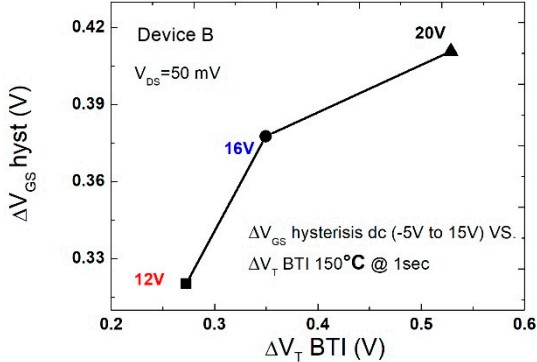

**Figure 12.** Mean values of $\Delta V_{th}$ induced by a positive bias temperature instability (PBTI) stress (at a fixed temperature, 150 °C and $t_{stress}$ = 1 s) and mean values of the hysteresis amplitude evaluated at ID = 5 mA against those measured.

This information will be useful to understanding how the devices switch dynamics and improve manufacturing, considering even the material and its quality. This information can be used for recommendations to manufacturers. SiC has more instability than silicon.

A positive threshold voltage shift reduces the performance in the on-state. Due to this consequence, the channel resistance of the devices is increased, which degrades the efficiency and the module temperature. To relax the BTI effect, it has to be made sure that devices which are used in a certain application show similarly narrow and predictable $\Delta V_{th}$ drifts. In the proposed case study, the devices showed the same trend of the threshold shift that ranged from 0.36 to 0.8 V in the experimental tests. The problem is a result of the interface and gate oxide quality of both devices.

The threshold hysteresis is a measure of the switching dynamics of the interface traps ($SiC/SiO_2$ interface or in the bulk $SiO_2$), while measurements at stress times (PBTI) can show the role of an additional interface degradation.

## 5. Conclusions

This paper presented the study of PBTI of SiC MOSFETs with $SiO_2$ and as the gate dielectric. For two commercially available SiC power MOSFETs, a positive $\Delta V_{th}$ is caused by an unification of interface trap generation and electron trapping in pre-existing oxide traps. The hysteresis amplitude exhibits a low sample-to-sample variability. Hysteresis amplitude at a fixed drain current is a function of the maximum applied gate voltage in Device A and Device B samples in pulsed conditions. Similar power law behavior was observed in the two similar types of SiC MOSFETs. Even if the devices from two distinct companies differ in the absolute values of the $\Delta V_{th}$, the identical tendencies of all traces show that all noted threshold voltage instabilities are commonly a fundamental physical property of the $SiC/SiO_2$ system and not associated with, e.g., mobile ion contamination. Most of the $\Delta V_{th}$ does not originate from an enduring deterioration of the interface, and is nearly fully recoverable.

Due to the applications of SiC, MOSFETs are not like logic integrated systems (IC) Si devices. Typical applications for SiC MOSFETs require high power densities and blocking voltages. The important part of this paper is the assessment of the impact and relevance of BTI for such applications. In this article we discussed how a BTI-induced threshold voltage shift could possibly affect the reliability and performance of such a high-power device. The positive shift of the threshold voltage reduces the overdrive in the on-state. Therefore, the channel resistance of single devices is increased, which degrades the efficiency.

**Author Contributions:** Formal analysis, E.G.; methodology, V.H.-P. and K.G.; software, C.R.; supervision, K.G.; validation, C.R.; writing—original draft preparation, E.G.; writing—review and editing, V.H.-P. All authors have read and agreed to the published version of the manuscript.

**Funding:** This research received no external funding.

**Conflicts of Interest:** The authors declare no conflict of interest.

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
