# Peer review of "Threshold Voltage Degradation for n-Channel 4H-SiC Power MOSFETs"

_jlpea, doi:10.3390/jlpea10010003_

Round 1

Reviewer 1 Report

The authors presented the study the variation of the threshold voltage of SiC power Mosfet using two commercially available devices.The study shows that all observed voltage instabilities are likely a fundamental physical property of the SiC/SiO2 system and not related to mobile ion contamination.

The manuscript has a lot of typos and grammar errors throughout the manuscript, which makes it sometimes very difficult to understand what is trying to be conveyed. The authors must correct them before it can be published in Journal of Low Power Electronics and Applications.

Author Response

Please, find in attached the file with the response.

Reviewer 2 Report

The authors have studied about threshold voltage stability of SiC (silicon carbide) MOSFET. The reviewer recommend this manuscript to publish in journal of Low Power Electronics and Application. However there are a few minor modifications required.

The reviewer thinks that the motivation of this research why the study of threshold voltage degradation is required should show in Abstract to help readers understand In Abstract, the authors talk about two characterization method but, there is unnecessary number at the end while they are concluding the abstract. Page2, line 49 and Page 4, line 1-4 MOSFET; Page 2, line 54 due to The reviewer recommends to use unified abbreviation for voltage threshold all over the manuscript which would prevent the readers’ confusion. The reviewer wonders if there is correlated result of One Spot Drop Down (OSDD) characterization method. This method is explained in the whole second chapter, but it would be helpful if the author shows any result or conclusion that correlate with the experimental process.

Author Response

(The authors gave the same response as above.)

Reviewer 3 Report

There is no detailed information of growth. In the case of commercionally available devices you don't have possibility to change anything. the changes just correspond to a tolerance of the product.

Author Response

(The authors gave the same response as above.)
